# Properties of Dietary Flavone Glycosides, Aglycones, and Metabolites on the Catalysis of Human Endoplasmic Reticulum Uridine Diphosphate Glucuronosyltransferase 2B7 (UGT2B7)

**DOI:** 10.3390/nu15234941

**Published:** 2023-11-28

**Authors:** Ting Xu, Yangjun Lv, Yuhan Cui, Dongchen Liu, Tao Xu, Baiyi Lu, Xuan Yang

**Affiliations:** 1College of Biosystems Engineering and Food Science, National-Local Joint Engineering Laboratory of Intelligent Food Technology and Equipment, Key Laboratory for Agro-Products Nutritional Evaluation of Ministry of Agriculture and Rural Affairs, Key Laboratory of Agro-Products Postharvest Handling of Ministry of Agriculture and Rural Affairs, Zhejiang Key Laboratory for Agro-Food Processing, Zhejiang International Scientific and Technological Cooperation Base of Health Food Manufacturing and Quality Control, Zhejiang University, Hangzhou 310058, China; 2Hangzhou Tea Research Institute, China Co-Op, Hangzhou 310016, China

**Keywords:** flavones, glycosylflavones, glycosides, phenolic acids, UGT2B7, nutrition utilization

## Abstract

Flavone glycosides, their aglycones, and metabolites are the major phytochemicals in dietary intake. However, there are still many unknowns about the cellular utilization and active sites of these natural products. Uridine diphosphate glucuronosyltransferases (UGTs) in the endoplasmic reticulum have gene polymorphism distribution in the population and widely mediate the absorption and metabolism of endogenous and exogenous compounds by catalyzing the covalent addition of glucuronic acid and various lipophilic chemicals. Firstly, we found that rutin, a typical flavone *O*-glycoside, has a stronger UGT2B7 binding effect than its metabolites. After testing a larger number of flavonoids with different aglycones, their aglycones, and metabolites, we demonstrated that typical dietary flavone *O*-glycosides generally have high binding affinities towards UGT2B7 protein, but the flavone *C*-glycosides and the phenolic acid metabolites of flavones had no significant effect on this. With the disposition of 4-methylumbelliferone examined by HPLC assay, we determined that 10 μM rutin and nicotifiorin could significantly inhibit the activity of recombinant UGT2B7 protein, which is stronger than isovitexin, vitexin, 3-hydroxyphenylacetic acid and 3,4-dihydroxyphenylacetic acid. In addition, in vitro experiments showed that in normal and doxorubicin-induced lipid composition, both flavone *O*-glycosides rutin and flavone *C*-glycosides isovitexin at 10 μM had no significant effect on the expression of *UGT1A1*, *UGT2B4*, *UGT2B7*, and *UGT2B15* genes for 24 h exposure. The obtained results enrich the regulatory properties of dietary flavone glycosides, aglycones, and metabolites towards the catalysis of UGTs and will contribute to the establishment of a precise nutritional intervention system based on lipid bilayers and theories of nutrients on endoplasmic reticulum and mitochondria communication.

## 1. Introduction

Flavone glycosides, their aglycones, and metabolites are commonly in foods of plant origin, major phytochemicals, and necessary for disaster risk reduction [1,2]. Intake of flavonols in the diet is important to improve lifespan through the regulation of the basic physiological metabolism [3,4] and mitochondria quality [5,6]. However, there are many forms of natural flavonoids and their derivatives, and there are still many unknown patterns of their absorption and metabolism within cells.

Compared to other types, flavonoid glycosides are the main form of human intake from plants. According to the types of glycosidic bonds, flavones can be divided into flavone *O*-glycosides and flavone *C*-glycosides. Although there have been studies comparing the differences in the performance of flavone *O*-glycides and flavone *C*-glycides in pyrolysis and enzymolysis [7], as well as the function in alleviating lipid accumulation [8,9], the mechanism by which this functional difference arises is still unclear.

Metabolism of the endogenous and exogenous compounds in human cells is mainly divided into two stages: Phase I metabolic reaction mainly involves redox reaction, and it hydrolyzes non-polar fat-soluble macromolecular substances into polar water-soluble small molecular metabolites. Phase II metabolic reaction combines the metabolites of phase I with endogenous small molecular organic substances for further utilization by the human body, which affects the utilization of nutrients and the outflow of toxins in human cells [10,11,12]. Phase II metabolic enzymes catalyze the binding reaction between metabolic intermediates generated by the phase I metabolic reaction [13,14,15]. In this process, glucu-ronosyltransferase catalyzes and mediates the binding reaction between the metabolic intermediates (including fatty acids, steroids, and glucuronic acid) and uridine diphosphate glucuronic acid (UDPGA) to generate glucuronic acid conjugates [16,17,18].

Uridine diphosphate glucuronosyltransferase (UGT) is a family of phase II binding enzymes that are widely distributed in various organs of the human body. It catalyzes the glucuronidation process and the covalent addition of glucuronic acid to generate a wide range of lipophilic chemicals, which is more conducive to the participation of substances in cell metabolism [15,19]. UGT includes two different families: UGT1 and UGT2 families. Most UGT1 families, like UGT1A1-A10, are involved in the metabolism of phenols and bilirubin. Most human UGT2 families include UGT2B4, UGT2B7, UGT2B10, UGT2B11, and UGT2B15, which are typically glycoside transferase transferring glucuronic acid moiety to hydroxyl, carboxyl, sulfhydryl, or amino groups of endogenous compounds and catalyzes the metabolism phytochemicals and fatty acids [20,21,22,23]. Clear evidence shows that UGTs are closely related to the efflux of chemicals and the formation of lipid bilayers [24]. The activity of UGTs can be influenced by the hydrophobic lipid environment in the ER membrane and the negative feedback regulated by membrane components, including phospholipids, cholesterol, and other proteins [25]. The current study shows that UGT2B is widely distributed in the biofilms of kidney cells, such as mitochondria and endoplasmic reticulum membranes [26]. UDP-Glucuronosyltransferase Family 2 Member B7 (UGT2B7) is the most common UGT2B member, located on the mitochondrial membranes, endoplasmic reticulum membranes, and the nuclear membranes of human cells, and it was widely studied because of the high correlation with metabolic diseases [27,28]. Previous studies noted that the activities of both UGT1A and UGT2B family members could be regulated by flavonols [29,30], but it is still unknown which form of dietary flavone glycosides, aglycones, or their metabolites play a leading role in this important topic.

4-methylumbelliferone (4-MU) has been widely used as a non-specific substrate of UGT to determine the effect of chemicals on UGT activity [30,31,32,33,34,35]. Catalyzed by UGT, 4-MU obtains the glucuronic acid group transferred from UDPGA through a glucuronidation binding reaction. Thus, 4-methyl-2-oxo-2H-1-benzopyran-7-yl-β-D-glucopyranosiduronic acid (4-MUG) is generated [36]. Here, we aimed to investigate the sites and effects of typical dietary flavone glycosides, aglycones, or their metabolites affecting human UGT enzymatic activity, thereby contributing to understanding the utilization patterns of flavonoids in cells.

## 2. Methods

### 2.1. Chemicals

Rutin, nicotifiorin, isovitexin, vitexin, HPAA, DHPAA, aucubin, and asperuloside were purchased from Yuanye Co. (Tianjin, China), with purity above 98% (HPLC). 4-MU, 4-MUG, 7-hydroxycoumarin, uridine-50-diphosphoglucuronic acid (UDPGA), and doxycycline (DOX) were purchased from Solarbio Co. (Beijing, China), with purity above 98% (HPLC). Recombined human UGT2B7 protein was obtained from IPHASE. All other commercially available reagents were of HPLC grade or the highest grade.

### 2.2. Gene Annotation Analysis of Uridine Diphosphate Glucuronosyltransferase 2B7

The chromosomal localization and tissue specificity analysis were obtained by GeneCards (https://www.genecards.org/, accessed on 6 June 2022). Human biological pathway unification was completed by PathCards (https://pathcards.genecards.org/, accessed on 6 June 2022). The diverse clinical and genetic annotation analysis of the human disease compendium was conducted by MalaCards (https://www.malacards.org/, accessed on 6 June 2022). Protein–protein interaction networks and the functional enrichment analysis were obtained using the STRING database (https://cn.string-db.org/, accessed on 6 June 2022).

### 2.3. Molecular Docking to Explain the Interaction of Flavone Glycosides, Phenolic Acids Metabolites, and Iridoids towards Uridine Diphosphate Glucuronosyltransferase 2B7 Activities

To compare the interaction between typical flavone glycosides, phenolic acids metabolites, and iridoids with UGT2B7, the molecular docking method was formed using SYBYL-X. The crystal structure of UGT2B7 protein (2O6L) was downloaded from the PDB database. Polar hydrogen atoms were added to the UGT isoform, non-polar hydrogen atoms were merged, and the UGT protein structural ends were repaired by terminal treatment. SDF files of phytochemical structures were downloaded from PubChem. Ligplot was used for the protein-fixed ligand-flexible docking calculations. With docking calculation, the ligands were calculated and ranked according to the total score, crash score, and polar energy terms.

### 2.4. HPLC Assessment of Inhibitory Effects of Flavones against Human Uridine Diphosphate Glucuronosyltransferase 2B7

According to the chromatogram of 4-MU, a nonselective substrate of UGT2B7, with different concentration gradients (0–1 mM), the standard curve of 4-MU under 316 nm irradiation in C18 HPLC column (4.6 × 250 mm, 5 μm, GL Science, Fukushima, Japan) at a flow rate of 1 mL/min was obtained [30,31,32,33,34,35]. Rutin, nicotifiorin, isovitexin, vitexin, HPAA, DHPAA, aucubin, and asperuloside were dissolved in DMSO. The total volume of 200 μL incubations contained 0.05 mg/mL recombinant human UGT2B7, 4 mM UDPGA, 5 mM MgCl_2_, 50 mM Tris HCl buffer (pH 7.4), and 1 mM 4-MU in the absence or presence of 1 μM–1 mM test substances in DMSO [37]. After a 5 min pre-incubation at 37 °C, the UDPGA was added to the mixture to initiate the reaction. The incubation time was 120 min. All incubations were terminated by adding 200 μL of acetonitrile containing internal standard (7-hydroxycoumarin, 100 μM). The incubation mixtures were then centrifuged at 12,000× *g* for 10 min at 4 °C. The products in the supernatant were measured by high-performance liquid chromatography (HPLC, Waters, Milford, MA, USA). The mobile phase consisted of acetonitrile (A) and 0.5% formic acid (B). All experiments were performed at least three times.

### 2.5. Cell Culture and Treatment

The human renal proximal tubular cell line (HKC) was purchased from the China Infrastructure of Cell Line Resource. Cell culture method according to previous studies [38], the culture medium was Dulbecco’s modified Eagle’s medium (DMEM) (Solarbio) supplemented with 10% fetal bovine serum (Evergreen, Taipei, Taiwan), 100 U/mL penicillin, 100 mg/mL streptomycin, 150 ng/mL amphotericin B (Macgene, Beijing, China), 2 mM L-Glutamine (Sigma, Livonia, MI, USA), and 1% MEM nonessential amino acids (Macgene) at 37 °C in 5% CO_2_ and 95% saturated atmospheric humidity. At approximately 70% confluence, the HKC cells were treated with 1 μM DOX, 10 μM rutin, isovitexin, and 1 μM DOX with these 50 μM phytochemicals for 24 h. Cells were transferred to ice before subsequent treatment.

### 2.6. Transcription Analysis of Flavones on Uuridine Diphosphate Glucuronosyltransferase Protein Expression

The total RNA of cells was isolated using Trizol reagent (Invitrogen, Carlsbad, CA, USA) and reverse transcribed into cDNA using the modified fastquant RT Kit (Tiangen, Beijing, China) method [39], followed by real-time quantitative PCR analysis (Thermofisher, Waltham, MA, USA) using SYBR Green (Sparkjade, Jinan, China). The primers used are listed in Table 1.

### 2.7. Statistical Analysis

The experimental data were shown as the mean value plus the standard deviation of three parallel experiments. Statistical analysis was carried out using GraphPad Prism 8.0. Multiple groups were subjected to a one-way analysis of variance (ANOVA), and the means were compared using a two-tailed unpaired Student’s *t*-test. Differences were considered to be significant for *p*-values less than 0.05.

## 3. Results

### 3.1. Overall Function of the Human Uridine Diphosphate Glucuronosyltransferase 2B7 

A combined tissue specificity analysis showed that the *UGT2B7* gene was widely expressed in metabolic organs (Figure 1A). Specifically, the kidney and liver are the two organs with the most abundant expression of *UGT2B7* in the human body. Whole blood, retina, heart, small intestines, adipocytes, lung, pancreas, and skin have a certain amount of *UGT2B7* expression, based on the Genecards database. However, the current results showed that the expression level of *UGT2B7* in both smooth muscle and skeletal muscle is limited, suggesting that the expression of *UGT2B7* is not only distributed in organs with vigorous metabolism but also in tissues with rich material exchange with diet and digestion. The pathway analysis result was carried out using the PathCards and is shown in Figure 1B. Multiple pharmacokinetics pathways, including the clomipramine pathway, statin pathway, zidovudine pathway, and bupropion pathway, are relevant to the function of *UGT2B7*. More interestingly, some metabolic pathways involving lipids and steroids are also enriched, such as estrogen metabolism, tamoxifen metabolism, and retinol metabolism. The correlation between abnormal expression of *UGT2B7* and human diseases was analyzed by MalaCards (Figure 1C). The cytochrome P450 oxidoreductase deficiency, breast cancer, bilirubin metabolic disorder, hepatocellular carcinoma, and Gilbert syndrome have the highest correlation with the abnormal expression of *UGT2B7*. In the human species, protein interaction analysis showed that cytochrome P450 monooxygenases are most directly related to UGT2B7 protein (Figure 1D). Gene ontology analysis of biological process enrichment showed that the function of UGT2B7 protein is closely related to positive regulation of DNA-directed DNA polymerase activity, detection of DNA damage, mitochondrial translational termination, and elongation. The top-rank molecular function enrichment is the glycogen synthase activity, structural constituent of ribosome, and nucleic acid binding. The top-rank cellular component enrichment is the eukaryotic translation initiation factor 3 complex, a component of the Replication factor C-like complex CTF18-RFC and mitochondrial small ribosomal subunit (Figure 1D).

### 3.2. Molecular Docking of Rutin, Its Phenolic Acid Metabolites, O-Glycosylflavones, and C-Glycosylflavones towards Uridine Diphosphate Glucuronosyltransferase 2B7 (2O6L)

The molecular docking of plant chemical ligands with UGT2B7 protein motifs (PDB ID: 2O6L) was used to explore the interaction and combination patterns between dietary flavone glycosides, aglycones, or their metabolites with UGT2B7. UGT2B7 is a nuclear gene located on human chromosome 4 (Figure 2A). Initially, we explored the preference for the combination pattern between UGT2B7 and rutin, a typical *O*-glycosylflavone, or its metabolites, including 3,4-dihydroxytoluene (DHT), 4-hydroxy-3-methoxyphenylacetic acid (HVA), 3,4-dihydroxyphenylacetic acid (DHPAA), and 3-hydroxyphenylacetic acid (HPAA) [40,41] (Figure 2B). According to the sequence alignment of GT1 family enzymes and the *C*-terminal domain sequences from human UGT2B7 [28], the amino acid residues of the binding capsule of UGT2B7 that bind to these biomolecules were summarized, including GLU291, MSE292, LEU353, LYS355, TRP356, ALA377, ASP401, ASN402, and HIS405 (Figure 2C). We found that rutin has a stronger binding capacity than its phenolic acid metabolites, which sparked our interest in whether there are differences between flavone glycosides, aglycones, and their metabolites in the regulation of UGT enzymatic activity, and their biological function may be one factor. The specific binding locations and the donor binding region are shown in Figure 2D,E. The hydrophobic contacts and involved amino acid residues are listed in Appendix A.

Continuing, we compared typical *O*-glycosylflavones, *C*-glycosylflavones rich in natural foods, and their binding force with UGT2B7. The results showed that the typical *O*-glycosylflavones have a stronger binding force with UGT2B7 than typical *C*-glycosylflavones. The specific binding sites of UGT2B7 with *O*-glycosylflavones and *C*-glycosylflavones are shown in Figure 3. The hydrophobic contacts and amino acid residues involved in this part are listed in Appendix A. Meanwhile, we also analyzed the aglycones of these flavone glycosides (Figure 4), and the hydrophobic contacts and involved amino acid residues are listed in Appendix A and Appendix A. Interestingly, the differential binding force between typical *O*-glycosylflavones and *C*-glycosylflavones to UGT2B7 did not emerge in their aglycones.

### 3.3. O-Glycosylflavone Compounds Generally Inhibit the Activity of Uridine Diphosphate Glucuronosyltransferase 2B7 

Glucuronidation reaction was used to evaluate the effects of *O*-glycosylflavones, *C*-glycosylflavones, their metabolites, and two other terpenoids on the activity of UGT2B7 protein. Based on the scoring results of molecular docking, rutin, nicotifiorin, isovitexin, vitexin, HPAA, DHPAA, aucubin, and asperuloside were chosen for comparison of effects. The HPLC assay for the UDPGA, 4-MU, and 4-MUG are shown in Figure 5A. The standard curve of 4-MU within 0–1 mM is shown in Figure 5B. We found that 10 μM rutin and nicotifiorin can significantly inhibit the activity of UGT2B7, and the IC50 scores were 74 μM and 98 μM, respectively (Figure 5C,D). However, the *C*-glycosylflavones and typical phenolic acid metabolites did not show this characteristic, and the activity of UGT was higher than 50% at 200 μM treatment (Figure 5E–H). In addition, the terpenoids, aucubin, and asperuloside, showed an inhibition effect of UGT2B7 activity at 10 μM, and the IC50 score was 60 μM and 100 μM, respectively (Figure 5I,J). The IC50 value is summarized in Appendix A.

### 3.4. Regulation of Flavone Glycosides on the Transcription Level of UGT Families in Normal and DOX-Induced Lipid Composition

Whether different types of flavone glycosides have an effect on the UGT gene expression is another topic worthy of attention. DOX can induce ER-mitochondria uncoupling, lipid-recycle blocking, and membrane lipid composition change [42]. Here, considering the molecular docking results, rutin and isovitexin, as typical representatives of flavonols with different glycosidic bonds, were used to evaluate their effect on UGT family expression in the HKC cells. In the normal group and 1 μM DOX-induced lipid composition, 10 μM rutin and isovitexin treatment with 24 h had no effect on the expression of *UGT1A1*, *UGT2B4*, *UGT2B7*, and *UGT2B15*, and there was no difference between the two flavonol glycosides groups (Figure 6A). On the other hand, 50 μM rutin and isovitexin with 24 h treatment reduced the expression of *UGT1A1*, *UGT2B4*, *UGT2B7*, and *UGT2B15* by 50%. *C*-glycosylflavones isovitexin had less inhibition on the expression of *UGT2B7* and *UGT2B15B* than rutin under normal conditions. To 1 μM DOX treatment, there was no significant difference in the expression of *UGT1A1*, *UGT2B4*, *UGT2B7*, and *UGT2B15* between rutin and isovitexin groups within 24 h (Figure 6B).

## 4. Discussion

Flavonols are known to have bioactivity to inhibit cyclooxygenase and lipoxygenase to control lipid metabolism and membrane structural integrity [43]. However, there are still large unknowns about the cellular utilization and active sites of these natural products. Based on the inspiration from rutin in our results, we believe that the roles of dietary flavone glycosides, aglycones, and their metabolites in regulating UGT enzymatic activity are different, which may be one of the direct factors for their different biofunctional potency. Flavone *O*-glycosides and flavone *C*-glycosides are two branches of flavones, but the utilization efficiency, tissue distributions, and metabolism of these substances are different [7,44]. The reason is not only related to the unbalanced distribution of blood vessels in tissues and the different distribution of transport receptors in membrane structures but also directly related to metabolic enzymes in organelles, especially in the endoplasmic reticulum. Through the glucuronidation reaction mediated by UGT enzymes, the liposolubility and metabolic characteristics of lipids, hormones, phenols, and carboxylic acids will undergo substantial morphological changes, affecting the outflow of chemical substances, the integrity of membrane structures, and even the homeostasis of the metabolic systems. We have paid attention to the UGT2B7 protein because of its polymorphism distribution in the population and high activity in metabolic organs such as the kidney and liver, which is also related to the utilization efficiency and transformation efficiency of nutrients in different human species [45,46]. Our results demonstrated that UGT2B7 was a locus responsible for the difference in bioavailability between flavone *O*-glycosides and flavone *C*-glycosides.

The phase II metabolic enzymes involved in the metabolism mainly include UGT, sulfotransferase (SULT), and catechol-O-methyltransferase (COMT) [47]. Among them, the UGT-catalyzed glucuronidation binding reaction is the most important metabolic pathway in human cells. However, the UGT may not be the only differential site. Flavones also have differential potential effects on the enzymatic activity of other Phase II metabolic enzymes, such as methyltransferases, phoryltransferases, and amino acid transferases. This is an important research direction for studying the absorption and utilization mechanism of flavones. We introduced iridoids because they have been proven to have a strong regulatory effect on lipophilic substance metabolism and human health [48,49]. We found that dietary iridoids have effects on UGT2B7 inhibition, which correlates with the reported role of glucosidase inhibitors [49]. Interestingly, the binding strength of flavone *C*-glycosides and phenolic acid metabolites to UGT2B7 is generally weak, which we further verified by HPLC assay. Phenolic acids are considered to have more extensive biological regulatory sites because their molecular weight is smaller than that of flavones, and they can cross the mitochondrial inner membrane to a certain extent [50]. Through comparison, we are more concerned that the inhibitory effect of *C*-glycosylflavones on UGT2B7 is lower than that of *O*-glycosylflavones in general. This may be an explanation and necessary supplement when administered flavone glycosides together with UGT substrate drugs [7,30]; the risk of *C*-glycosylflavones is relatively lower than that of *O*-glycosylflavones. On the other hand, it is worth noting that improving the loading performance has always been a key problem in the cell-based drug delivery system [51]. The general inhibition of flavone *O*-glycosides on UGT2B7 is a potential strategy for the application of natural products to regulate drug efflux by inhibiting glucuronidation reaction, thus increasing the loading performance.

To further compare the effects on metabolism in living cells, we focused on the correlation between the dose effect of flavone glycosides and the expression level of UGT enzyme in a human renal tubular epithelial cell line, where there exists large endoplasmic reticulum membranous surfaces, an active platform of chemicals exchange and also an important model to study the ER homeostasis and inflammation [52]. The results pointed out that low concentrations of flavone *O*-glycosides and flavone *C*-glycosides treatment did not affect the expression of UGT at the transcriptional level. However, 50 μM rutin and isovitexin showed a disturbing inhibition of UGT, suggesting a potential risk of lipid metabolism regulation. DOX-induced lipid composition is based on UGT inhibition and the over-production of peroxynitrite that induces endoplasmic reticulum disorder and mitochondrial damage [53,54,55]. Thus, we introduced it as the lipid process-blocking group caused by UGT inhibition. It was found that although flavone *O*-glycosides inhibited UGT activity, it did not further reduce DOX-mediated UGT expression.

In recent years, a large number of foods and supplements with the lipid bilayer and mitochondrial health regulatory effects to reduce the risk of disease have been reported, including flavone *O*-glycosides and flavone *C*-glycosides components [56]. Currently, there are studies that have confirmed the interaction between *O*-glycosides and UGT2B7 [57], but we have not found any report on the interaction between *C*-glycosides and UGT2B7. However, the sites and applicability of these components in cell metabolism have not been well clarified. The upstream interaction sites of UGT in human cells are the phase I metabolic enzymes cytochrome P450 (CYP) [58,59], which localize in the ER membrane, regulate the ER-mitochondria communication [60], catalyze the metabolism of exogenous compounds, and mediate the lipophilic substance synthesis [61]. Notably, UGT genetic variations were polymorphically distributed across different populations, converging in Asia, Europe, and Africa, which also indicates differences in their utilization of nutrients [62,63,64,65]. Considering that flavone molecules cannot directly enter the mitochondrial inner membranes, the regulation of flavones on the UGT2B7 polymorphism population could affect the composition of organelle membranes and support the establishment of a precise nutritional intervention system based on lipid bilayers.

## 5. Conclusions

In conclusion, our study shows that typical dietary flavone *O*-glycosides generally have a strong binding with UGT2B7 protein, but the flavone *C*-glycosides, their aglycones, and metabolites have no significant effect on this, which we summarized in Figure 7. We determined that 10 μM rutin and nicotifiorin could significantly inhibit the activity of the UGT2B7 protein, which is stronger than isovitexin, vitexin, HPAA, and DHPAA. In vitro experiments showed that 10 μM flavone *O*-glycosides rutin and flavone *C*-glycosides isovitexin had no significant effect on the expression of *UGT1A1, UGT2B4, UGT2B7*, and *UGT2B15* genes in renal tubular epithelial cells. This study focused on the bioavailability of flavones and found that UGT2B7-mediated glucuronidation of flavone *O*-glycosides was generally stronger than that of flavone *C*-glycosides and phenolic acids. This study enriches the regulatory properties of dietary flavone glycosides, aglycones, and metabolites on the catalysis of UGTs and will contribute to the establishment of precise nutritional intervention systems based on lipid bilayers and theories of nutrient utilization on endoplasmic reticulum and mitochondria.

## Figures and Tables

**Figure 1 nutrients-15-04941-f001:**
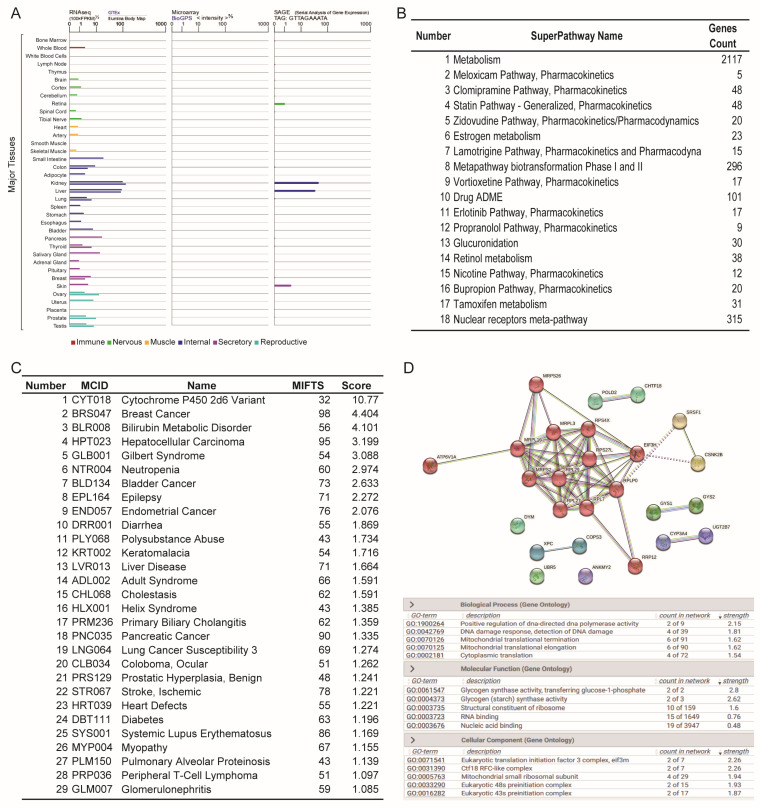
**Tissue specificity and functional analysis of *UGT2B7* gene expression.** (**A**) *UGT2B7* mRNA expression in normal human tissues from GTEx, Illumina, BioGPS, and SAGE, data from GeneCards. (**B**) Pathway unification of *UGT2B7* gene by PathCards analysis. (**C**) Analysis of its association with human diseases by MalaCards analysis (Score above 1.000). (**D**) Gene association and gene ontology analysis by STRING database.

**Figure 2 nutrients-15-04941-f002:**
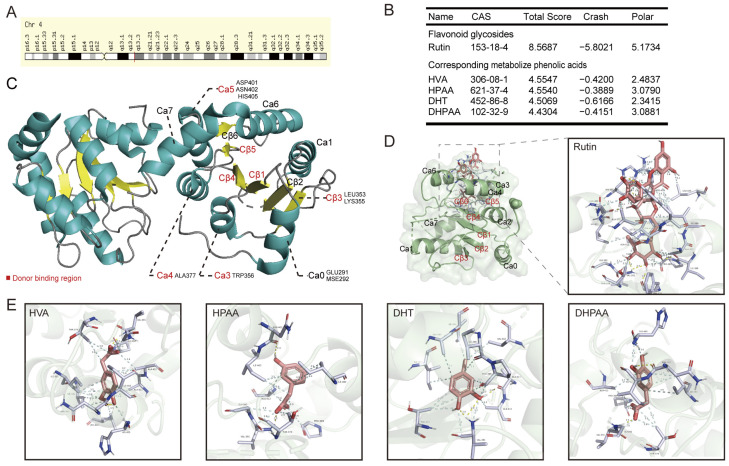
**Molecular docking of typical *O*-glycosylflavones rutin and its phenolic acid metabolites towards UGT2B7 (2O6L).** (**A**) The chromosomal localization of UGT2B7 protein. (**B**) The preference of combination pattern between the typical *O*-glycosylflavone rutin and its four metabolized phenolic acids with UGT2B7. (**C**) Summary of specific binding locations between these phytochemical ligands and the complete recombinant UGT2B7 chain A. Donor binding region of UGT2B7 was colored in red, the alpha helix area was colored in green, and the beta folding area was colored in yellow. Stereo views of Ligplot algorithm model of (**D**) rutin, (**E**) phenolic acid metabolites of rutin -UGT2B7 protein interactions.

**Figure 3 nutrients-15-04941-f003:**
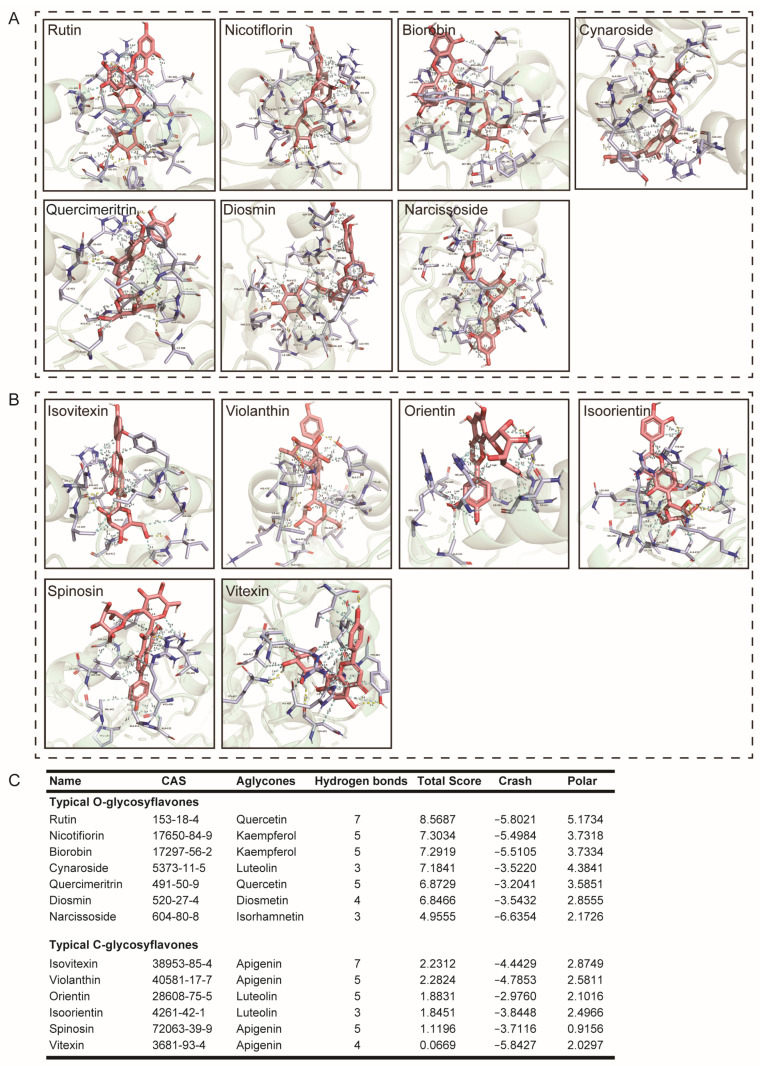
**Molecular docking of typical *O*-glycosylflavones and *C*-glycosylflavones towards UGT2B7 (2O6L).** Stereo views of Ligplot algorithm model of (**A**) *O*-glycosylflavones, (**B**) *C*-glycosylflavones-UGT2B7 protein interactions. (**C**) The preference of combination pattern between seven typical *O*-glycosylflavones, six typical *C*-glycosylflavones, and activity cavity of typical UGT2B7.

**Figure 4 nutrients-15-04941-f004:**
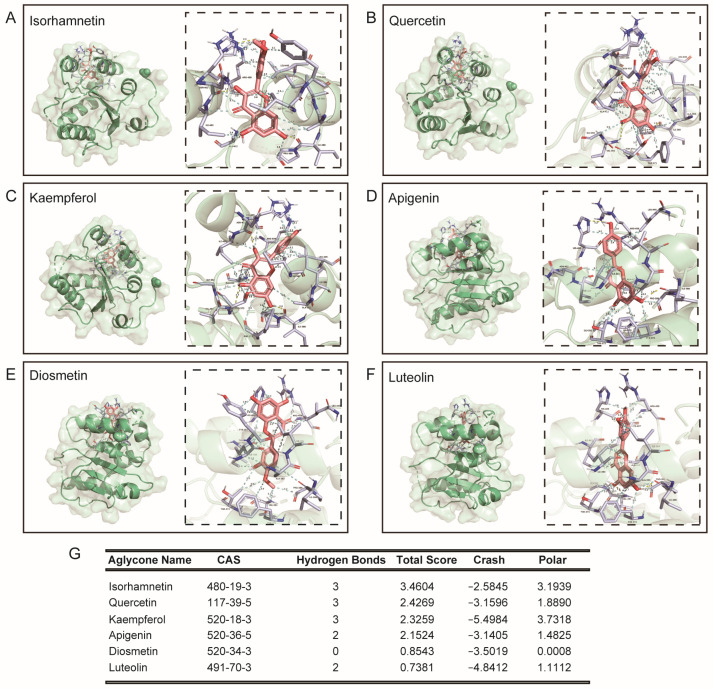
**Molecular docking of typical flavonols towards UGT2B7 (2O6L).** Stereo views of Ligplot algorithm model of (**A**) isorhamnetin, (**B**) quercetin, (**C**) kaempferol, (**D**) apigenin, (**E**) diosmetin and (**F**) luteolin-UGT2B7 protein interactions. (**G**) The preference of combination pattern between the six typical flavonols and activity cavity of typical UGT2B7.

**Figure 5 nutrients-15-04941-f005:**
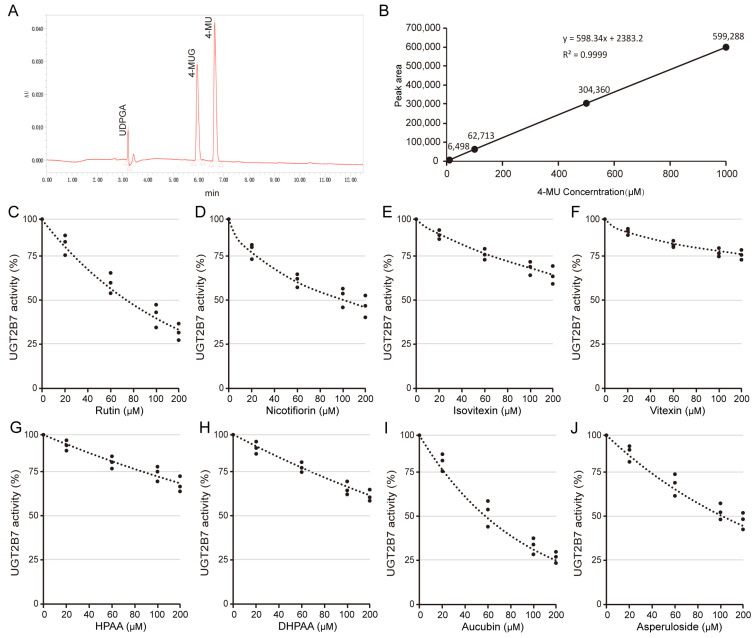
**Dose-dependent inhibition behavior of typical flavone glycosides and other components toward recombinant human UGT2B7.** (**A**) A direct HPLC assay for the determination of UDPGA, 4-MU, and 4-MUG. (**B**) The standard curve of 0–1 mM 4-MU. Inhibition screening of UGT2B7 by (**C**) rutin, (**D**) nicotifiorin, (**E**) isovitexin, (**F**) vitexin, (**G**) HPAA, (**H**) DHPAA, (**I**) aucubin, and (**J**) asperuloside, respectively. Data are shown using mean values plus S.D. (*n* = 3).

**Figure 6 nutrients-15-04941-f006:**
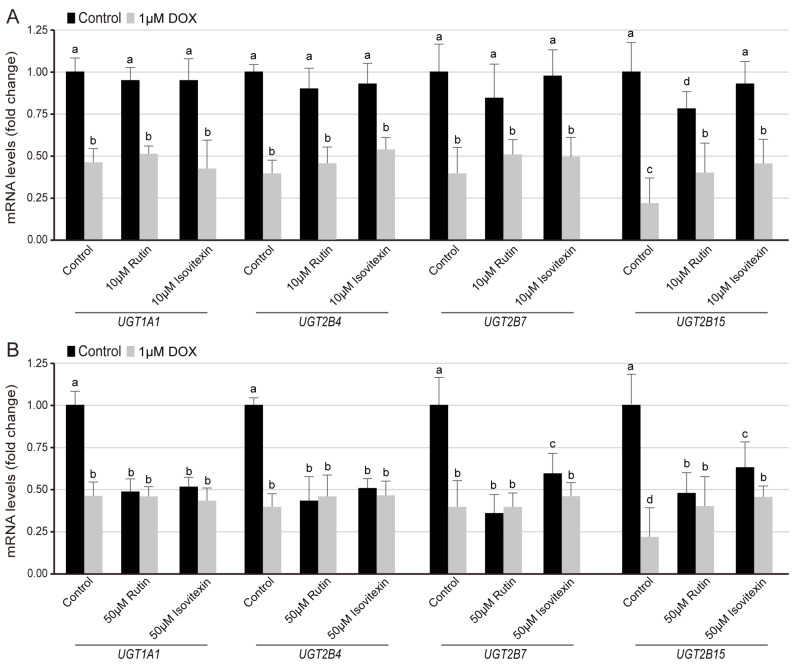
**Effect of rutin and isovitexin on the transcription level of the UGT families.** QRT-PCR analysis of the relative expression of *UGT1A1*, *UGT2B4*, *UGT2B7*, and *UGT2B15* genes compared with GAPDH in HKC cells for 24 h under control and DOX conditions. (**A**) Exposure to the 10 μM rutin or isovitexin, (**B**) Exposure to the 50 μM rutin or isovitexin. Data acquisition used more than 1 × 10^7^ cells. All the data were expressed as a percentage of the control, presented as the means ± SD of three independent experiments. The data were analyzed using one-way ANOVA. Different characters indicate significant differences between the compared groups (*p* < 0.05).

**Figure 7 nutrients-15-04941-f007:**
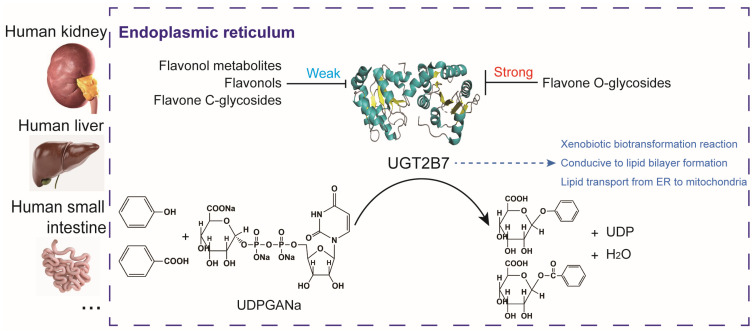
**Schematic diagram of the regulation of UGT2B7 by typical flavonol derivatives.** UGT2B7 is an important enzyme that catalyzes the glucuronidation of phenols, benzoic acids, and other substances. It regulates xenobiotic biotransformation reaction and intracellular transport of lipid molecules and widely spreads in the endoplasmic reticulum of human kidney, liver, small intestine, and other tissues. Glucuronidation also enhances the conductivity of lipid bilayer formation. Typical flavone *O*-glycosides generally inhibit the catalytic efficiency of UGT2B7, while typical flavonol metabolites, flavonols, and flavone *C*-glycosides show weak inhibition effects on UGT2B7, suggesting the differences in the regulation of UGT2B7 activity by forms of flavonoid derivatives. Organ images are copyrighted from https://www.turbosquid.com (Accessed on 8 August 2022).

**Table 1 nutrients-15-04941-t001:** Primers used in the quantitative real-time PCR assays.

Primers	Sequence (5′→3′)
*UGT1A1-F*	*GCTTTTGTCTGGCTGTTCCCACT*
*UGT1A1-R*	*TCGAAGGTCATGTGATCTGAATGAGA*
*UGT2B4-F*	*GTATTGGCATCTTCAGCTTCCATTTC*
*UGT2B4-R*	*AAGTTCTGCCCATCTCTTAACCAGC*
*UGT2B7-F*	*TTTCACAAGTACAGGAAATCATGTCAAT*
*UGT2B7-R*	*CAGCAGCTCACTACAGGGAAAAAT*
*UGT2B15-F*	*GAAAATTCTCGATAGATGGATATATGGTG*
*UGT2B15-R*	*AACTGCATCTTTACAGAGCTTGTTACTG*

## Data Availability

Data are contained within the article and Appendix A.

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
