# Peer review of "Properties of Dietary Flavone Glycosides, Aglycones, and Metabolites on the Catalysis of Human Endoplasmic Reticulum Uridine Diphosphate Glucuronosyltransferase 2B7 (UGT2B7)"

_nutrients, 2023, doi:10.3390/nu15234941_

Round 1

Reviewer 1 Report

Comments and Suggestions for Authors

Dear Authors,

It is a very well-executed project. Phase II metabolite glucuronic acid ones are polar. Glucuronic acid makes them water soluble and they get excreted via urinary tract. The xenobiotic transformation is a very important process of xenobiotic elimination from the body. Endo plasmic reticulum is an important organelle to perform these phase II biotransformations.

Flavone - Ol glycosides are strong metabolites and substrates for UGT287. The dose response metabolism is very useful for predicting this metabolism for a similar xenobiotic. But, what happens to other Phase II metabolizing enzymes like:

Methy transferases

Phoryl transferases

Aminoacid Transferases

Glutothione transferases

Does these Phase II enzyme macinery is reduced in action? Any comments on this line?

Enjoyed reading it.

Comments on the Quality of English Language

English editing with minor changes will elevate.

Reviewer 2 Report

Comments and Suggestions for Authors

Firstly, there is a lack of logical coherence in the experimental design overall, and the description of experimental details and results is unclear. Most importantly, there are numerous errors to the extent that sentences are difficult to comprehend. Major points are detailed below.

1.     The logic presented by the authors in the submitted paper was difficult to follow. It is necessary to improve the overall logical structure and grammatical aspects of the paper.

2.     It is not clear which 4-MU (the substrate) or 4-MUG (the metabolite) was quantified when evaluating UGT activity. Moreover, the rationale behind presenting 4-MUG in the chromatogram as quantifiable while showing only a calibration curve with 4-MU is not explained.

3.     The criteria for selecting specific compounds for evaluating UGT inhibitory activity and how these choices relate to the results from the docking model are unclear.

4.     The choice of using renal epithelial cells instead of liver or intestinal cells, where UGT metabolism predominantly occurs, for studying UGT expression changes is not explained. Additionally, the basis for determining substrate concentrations in these experiments is not provided.

5.     The significance of flavonoid treatment in the Doxorubicin-treated group is not elaborated upon.

Comments on the Quality of English Language

There are many grammatically incorrect parts, unclear expressions, and strange sentence structures, making it difficult to comprehend the content.

Reviewer 3 Report

Comments and Suggestions for Authors

The main question addressed by the research group is “the catalytic effect of various flavone glycosides, aglycones, and their metabolites on human endoplasmic reticulum uridine diphosphate glucuronosyltransferase UGT2B7”. The topic is original and very relevant to the field and discussed novel aspects of the metabolic potential of uridine diphosphate glucuronosyltransferase UGT2B7.” The authors have paved the way for future investigations of the metabolic crosstalk between endoplasmic reticulum and mitochondria. The methodology is well designed, and in case authors want to investigate in greater depth, advanced 3D cell culture models or extensive in vivo experimentation can be carried out.

·      The manuscript is written well, and the results are explained comprehensively.

·      The authors can improve the title of the manuscript to make it more specific to their study.

·      Refer to line 23, the authors should precisely provide the name of the testing method to clarify any doubts in the reader’s mind.

·      Authors should consider adding a graphical abstract for their study.

Comments on the Quality of English Language

 Moderate editing of English language is required.

Author Response

Response to reviewers

The manuscript is written well, and the results are explained comprehensively.

  1. The authors can improve the title of the manuscript to make it more specific to their study.

Response:

Thank you for your suggestion.The title has now been adjusted to “Properties of dietary flavone glycosides, aglycones, and metabolites on the catalysis of human endoplasmic reticulum uridine diphosphate glucuronosyltransferase 2B7 (UGT2B7)”

  1. Refer to line 23, the authors should precisely provide the name of the testing method to clarify any doubts in the reader’s mind.Authors should consider adding a graphical abstract for their study.

Response:

Thank you for your suggestion.Wehave changed the title:

Line 122 from “Assessment of inhibitory effects of flavones against human UGT2B7” to “HPLC assessment of inhibitory effects of flavones against human UGT2B7”

Line147 from “UGTs gene expression analysis” to “Transcription analysis of flavones on UGT protein expression”

Thanks, we used the Figure7 Schematic diagram of the regulatory of typical flavonol derivatives on UGT2B7, as the graphical abstract.

Reviewer 4 Report

Comments and Suggestions for Authors

The manuscript by Xu et al reports on the regulatory properties of dietary flavone glycosides, aglycones, and metabolites on the catalysis of UGTs. Based on the docking results, Xu et al found that flavone O-glycosides generally have a stronger binding with UGT2B7 protein than C-glycosides. In my evaluation, I found the manuscript to be well-organized, with the experiments and calculations conducted diligently. However, I do have a suggestion for the authors to further enhance their work.

1)     It would be beneficial if they made a more concerted effort to provide some more details of simulations in the supplementary file. Authors should add more information about the results of the molecular docking calculations like an image of the best docking poses for each compound etc.

2)     The author measured levels of mRNA of UGT2B7, however, the mRNA does not translate to protein expression always. It would be good if the author also can provide protein expression data to further support their data.

3)     Please, provide a tabulated summary of IC50 values of figure 4 data.

4)     Please, report if there are key structural features of C- vs O-glycosides that provide binding interactions with the UGT2B7 active site. Also reports which amino acids at the UGT2B7 active site provide binding interactions with glycosides.

5)     Please, cite any examples of C-/O-glycosides-drug interactions of UGT2B7.

6)     Also, please explain why alamethicin was not added to the UGT activity assay.

7)     Please, add appropriate references wherever required. For example, lines 160-164 references are missing in the text.

Comments on the Quality of English Language

None

Round 2

Reviewer 2 Report

Comments and Suggestions for Authors

Ambiguous and difficult-to-understand sentences are found in several places, and grammatical corrections need to be made.

1.     The title seems inadequate in capturing the content of the paper, so it appears that a revision is necessary.

2.     Abstract line 26; Does “With confirmed by HPLC assay,” means “Inhibitory effects of flavones on 4-MU metabolism by recombinant UGT2B7 enzymes”?

3.     Abstract line 29; What is the “lipid composition condition”? Does it mean “lipid accumulation state”?

4.     Page 2 line 80-83; “Previous studies noted that the activities of both UGT1A and UGT2B family members can be mediated by flavonols [29,30], but it is still unknown which form of dietary flavone glycosides, aglycones, or metabolites play a leading role in this important topic.“

; I cannot understand why UGTs activities are mediated by flavonols?

5.     Pate 2 line 90-91; What is the “flavone-mediated metabolic characteristics”?

6.     Page 17 figure 5 legend; “typically” should be “typical”.

7.     I haven't mentioned everything, but there are too many unclear or ambiguous expressions, as well as incorrect ones, in the manuscript.. I believe that meticulous proofreading of the manuscript for English corrections and clarification of technical content is still necessary.

8.     You concluded that “both flavone O-glycosides rutin and flavone C-glycosides isovitexin had no significant effect on the expression of UGT1A1, UGT2B4, UGT2B7, and UGT2B15 genes.” However, high concentrations of rutin and isovitexin did have some effects on the expression of UGTs in normal cells.

Comments on the Quality of English Language

I believe that meticulous proofreading of the manuscript for English corrections and clarification of technical content is still necessary. 
